# Efficient Marginalization of
# Discrete and Structured Latent Variables via Sparsity

**Gonçalo M. Correia**[Ω]
goncalo.correia@lx.it.pt

**Vlad Niculae**[Ω*]
vlad@vene.ro

**Wilker Aziz**[M]
w.aziz@uva.nl

**André F. T. Martins**[Ω Ψ ℮]
andre.t.martins@tecnico.ulisboa.pt

[Ω]Instituto de Telecomunicações, Lisbon, Portugal
[Ψ]LUMLIS (Lisbon ELLIS Unit), Instituto Superior Técnico, Lisbon, Portugal
[℮]Unbabel, Lisbon, Portugal
[M]ILLC, University of Amsterdam, The Netherlands
[Ω]IvI, University of Amsterdam, The Netherlands

## Abstract

Training neural network models with discrete (categorical or structured) latent variables can be computationally challenging, due to the need for marginalization over large or combinatorial sets. To circumvent this issue, one typically resorts to sampling-based approximations of the true marginal, requiring noisy gradient estimators (e.g., score function estimator) or continuous relaxations with lower-variance reparameterized gradients (e.g., Gumbel-Softmax). In this paper, we propose a new training strategy which replaces these estimators by an exact yet efficient marginalization. To achieve this, we parameterize discrete distributions over latent assignments using differentiable sparse mappings: sparsemax and its structured counterparts. In effect, the support of these distributions is greatly reduced, which enables efficient marginalization. We report successful results in three tasks covering a range of latent variable modeling applications: a semisupervised deep generative model, a latent communication game, and a generative model with a bit-vector latent representation. In all cases, we obtain good performance while still achieving the practicality of sampling-based approximations.

## 1 Introduction

Neural latent variable models are powerful and expressive tools for finding patterns in high-dimensional data, such as images or text [1–3]. Of particular interest are *discrete* latent variables, which can recover categorical and structured encodings of hidden aspects of the data, leading to compact representations and, in some cases, superior explanatory power [4, 5]. However, with discrete variables, training can become challenging, due to the need to compute a gradient of a large sum over all possible latent variable assignments, with each term itself being potentially expensive. This challenge is typically tackled by estimating the gradient with Monte Carlo methods [MC; 6], which rely on sampling estimates. The two most common strategies for MC gradient estimation are the score function estimator [SFE; 7, 8], which suffers from high variance, or surrogate methods that rely on the continuous relaxation of the latent variable, like straight-through [9] or Gumbel-Softmax [10, 11] which potentially reduce variance but introduce bias and modeling assumptions.

In this work, we take a step back and ask: Can we avoid sampling entirely, and instead deterministically evaluate the sum with less computation? To answer affirmatively, we propose an alternative method to train these models by parameterizing the discrete distribution with **sparse mappings** — sparsemax [12] and two structured counterparts, SparseMAP [13] and a novel mapping top-$k$ sparsemax. Sparsity implies that some assignments of the latent variable are entirely ruled out. This leads to the corresponding terms in the sum evaluating trivially to zero, allowing us to disregard potentially expensive computations.

**Contributions.** We introduce a general strategy for learning deep models with discrete latent variables that hinges on learning a sparse distribution over the possible assignments. In the unstructured categorical case our strategy relies on the sparsemax activation function, presented in §3, while in the structured case we propose two strategies, SparseMAP and top-$k$ sparsemax, presented in §4. Unlike existing approaches, our strategies involve neither MC estimation nor any relaxation of the discrete latent variable to the continuous space. We demonstrate our strategy on three different applications: a semisupervised generative model, an emergent communication game, and a bit-vector variational autoencoder. We provide a thorough analysis and comparison to MC methods, and — when feasible — to exact marginalization. Our approach is consistently a top performer, combining the accuracy and robustness of exact marginalization with the efficiency of single-sample estimators.[2]

**Notation.** We denote scalars, vectors, matrices, and sets as $a$, $\boldsymbol{a}$, $\boldsymbol{A}$, and $\mathcal{A}$, respectively. The indicator vector is denoted by $\boldsymbol{e}_i$, for which every entry is zero, except the $i^{\text{th}}$, which is 1. The simplex is denoted $\triangle^K := \{\boldsymbol{\xi} \in \mathbb{R}^K : \langle \mathbf{1}, \boldsymbol{\xi} \rangle = 1, \boldsymbol{\xi} \geq \mathbf{0}\}$. $\mathbb{H}(p)$ denotes the Shannon entropy of a distribution $p(z)$, and $\mathsf{KL}\,[p||q]$ denotes the Kullback-Leibler divergence of $p(z)$ from $q(z)$. The number of non-zeros of a sequence $z$ is denoted $\|z\|_0 := |\{t : z_t \neq 0\}|$. Letting $z \in \mathcal{Z}$ be a random variable, we write the expectation of a function $f : \mathcal{Z} \to \mathbb{R}$ under distribution $p(z)$ as $\mathbb{E}_{p(z)}[f(z)]$.

## 2 Background

We assume throughout a latent variable model with observed variables $x \in \mathcal{X}$ and latent stochastic variables $z \in \mathcal{Z}$. The overall fit to a dataset $\mathcal{D}$ is $\sum_{x \in \mathcal{D}} \mathcal{L}_x(\theta)$, where the loss of each observation,

$$\mathcal{L}_x(\theta) = \mathbb{E}_{\pi(z|x,\theta)}\,[\ell(x, z; \theta)] = \sum_{z \in \mathcal{Z}} \pi(z|x, \theta)\, \ell(x, z; \theta)\,, \qquad (1)$$

is the expected value of a downstream loss $\ell(x, z; \theta)$ under a probability model $\pi(z|x, \theta)$ of the latent variable; in other words, the latent variable $z$ is *marginalized* to compute this loss. To model complex data, one parameterizes both the downstream loss and the distribution over latent assignments using neural networks, due to their flexibility and capacity [2].

In this work, we study **discrete** latent variables, where $|\mathcal{Z}|$ is finite, but possibly very large. One example is when $\pi(z|x, \theta)$ is a categorical distribution, parameterized by a vector $\boldsymbol{\xi} \in \triangle^{|\mathcal{Z}|}$. To obtain $\boldsymbol{\xi}$, a neural network computes a vector of scores $\boldsymbol{s} \in \mathbb{R}^{|\mathcal{Z}|}$, one score for each assignment, which is then mapped to the probability simplex, typically via $\boldsymbol{\xi} = \mathsf{softmax}(\boldsymbol{s})$. Another example is when $\mathcal{Z}$ is a structured (combinatorial) set, such as $\mathcal{Z} = \{0, 1\}^D$. In this case, $|\mathcal{Z}|$ grows exponentially with $D$ and it is infeasible to enumerate and score all possible assignments. For this structured case, scoring assignments involves a decomposition into parts, which we describe in §4.

Training such models requires summing the contributions of all assignments of the latent variable, which involves as many as $|\mathcal{Z}|$ evaluations of the downstream loss. When $\mathcal{Z}$ is not too large, the expectation may be evaluated explicitly, and learning can proceed with exact gradient updates. If $\mathcal{Z}$ is large, and/or if $\ell$ is an expensive computation, evaluating the expectation becomes prohibitive. In such cases, practitioners typically turn to MC estimates of $\nabla_\theta \mathcal{L}_x(\theta)$ derived from latent assignments sampled from $\pi(z|x, \theta)$. Under an appropriate learning rate schedule, this procedure converges to a local optimum of $\mathcal{L}_x(\theta)$ as long as gradient estimates are unbiased [14]. Next, we describe the two current main strategies for MC estimation of this gradient. Later, in §3–4, we propose our **deterministic** alternative, based on sparsifying $\pi(z|x, \theta)$.

**Monte Carlo gradient estimates.** Let $\theta = (\theta_\pi, \theta_\ell)$, where $\theta_\pi$ is the subset of weights that $\pi$ depends on, and $\theta_\ell$ the subset of weights that $\ell$ depends on. Given a sample $z \sim \pi(z|x, \theta_\pi)$, an unbiased estimator of the gradient for Eq. 1 *w.r.t.* $\theta_\ell$ is $\nabla_{\theta_\ell} \mathcal{L}_x(\theta) \approx \nabla_{\theta_\ell} \ell(x, z; \theta_\ell)$. Unbiased estimation of $\nabla_{\theta_\pi} \mathcal{L}_x(\theta)$ is more difficult, since $\theta_\pi$ is involved in the sampling of $z$, but can be done with SFE [7, 8]: $\nabla_{\theta_\pi} \mathcal{L}_x(\theta) \approx \ell(x, z; \theta_\ell) \nabla_{\theta_\pi} \log \pi(z|x, \theta_\pi)$, also known as REINFORCE [15]. The SFE is powerful and general, making no assumptions on the form of $z$ or $\ell$, requiring only a sampling oracle and a way to assess gradients of $\log \pi(z|x, \theta_\pi)$. However, it comes with the cost of high variance. Making the estimator practically useful requires variance reduction techniques such as baselines [15, 16] and control variates [17–19]. Variance reduction can also be achieved with Rao-Blackwellization techniques such as sum and sample [20–22], which marginalizes an expectation over the top-$k$ elements of $\pi(z|x, \theta_\pi)$ and takes a sample estimate from the complement set.

**Reparameterization trick.** For continuous latent variables, low-variance pathwise gradient estimators can be obtained by separating the source of stochasticity from the sampling parameters, using the so-called *reparameterization trick* [2, 3]. For discrete latent variables, reparameterizations can only be obtained by introducing a step function like argmax, which has null gradients almost everywhere. Replacing the gradient of argmax with a nonzero surrogate like the identity function, known as Straight-Through [9], or with the gradient of softmax, known as *Gumbel-Softmax* [10, 11], leads to a biased estimator that can still perform well in practice. Continuous relaxations like Straight-Through and Gumbel-Softmax are only possible under a further modeling assumption that $\ell$ is defined continuously (thus differentiably) in a neighbourhood of the indicator vector $\boldsymbol{z} = \boldsymbol{e}_z$ for every $z \in \mathcal{Z}$. In contrast, both SFE-based methods as well as our approach make no such assumption.

## 3 Efficient Marginalization via Sparsity

The challenge of computing the exact expectation in Eq. 1 is linked to the need to compute a sum with a large number of terms. This holds when the probability distribution over latent assignments is *dense* (*i.e.*, every assignment $z \in \mathcal{Z}$ has non-zero probability), which is indeed the case for most parameterizations of discrete distributions. Our proposed methods hinge on *sparsifying* this sum.

Take the example where $\mathcal{Z} = \{1, \ldots, K\}$, with a neural network predicting from $x$ a $K$-dimensional vector of real-valued scores $\boldsymbol{s} = \boldsymbol{g}(x; \theta)$, such that $s_z$ is the score of $z$.[3] The traditional way to obtain the vector $\boldsymbol{\xi}$ parameterizing $\pi(z|x, \theta)$ is with the softmax transform, *i.e.* $\boldsymbol{\xi} = \mathsf{softmax}(\boldsymbol{s})$. Since this gives $\pi(z|x, \theta) \propto \exp(s_z)$, the expectation in Eq. 1 depends on $\ell(x, z; \theta)$ for every possible $z$.

We rethink this standard parameterization, proposing a **sparse** mapping from scores to the simplex. In particular, we substitute softmax by sparsemax [12]:

$$\mathsf{sparsemax}(\boldsymbol{s}) \coloneqq \underset{\boldsymbol{\xi} \in \triangle^K}{\arg\min} \|\boldsymbol{\xi} - \boldsymbol{s}\|_2^2 . \tag{2}$$

Like softmax, sparsemax is differentiable and has efficient forward and backward passes [23, 12], described in detail in App. A; the backward pass is essential in our use case. Since Eq. 2 is the Euclidean projection operator onto the probability simplex, and solutions can hit the boundary, sparsemax is likely to assign **probabilities of exactly zero**; in contrast, softmax is always dense.

Our main insight is that with a sparse parameterization of $\pi$, we can compute the expectation in Eq. 1 evaluating $\ell(x, z; \theta)$ only for assignments $z \in \bar{\mathcal{Z}} \coloneqq \{z : \pi(z|x, \theta) > 0\}$. This leads to a powerful alternative to MC estimation, which requires fewer than $|\mathcal{Z}|$ evaluations of $\ell$, and which strategically — yet deterministically — selects which assignments $\bar{\mathcal{Z}}$ to evaluate $\ell$ on. Empirically, our analysis in §5 reveals an adaptive behavior of this sparsity-inducing mechanism, performing more loss evaluations in early iterations while the model is uncertain, and quickly reducing the number of evaluations, especially for unambiguous data points. This is a notable property of our learning strategy: In contrast, MC estimation cannot decide when an ambiguous data point may require more sampling for accurate estimation; and directly evaluating Eq. 1 with the dense $\boldsymbol{\xi}$ resulting from a softmax parameterization never reduces the number of evaluations required, even for simple instances.

# 4 Structured Latent Variables

While the approach described in §3 theoretically applies to any discrete distribution, many models of interest involve structured (or combinatorial) latent variables. In this section, we assume $z$ can be represented as a *bit-vector—i.e.* a random vector of discrete binary variables $\boldsymbol{a}_z \in \{0, 1\}^D$. This assignment of binary variables may involve global factors and constraints (*e.g.* tree constraints, or budget constraints on the number of active variables, *i.e.* $\sum_i [\boldsymbol{a}_z]_i \leq B$, where $B$ is the maximum number of variables allowed to activate at the same time). In such structured problems, $|\mathcal{Z}|$ increases exponentially with $D$, making exact evaluation of $\ell(x, z; \theta)$ prohibitive, even with sparsemax.

Structured prediction typically handles this combinatorial explosion by parameterizing scores for individual binary variables and interactions within the global structured configuration, yielding a compact vector of **variable scores** $\boldsymbol{t} = \boldsymbol{g}(x; \theta) \in \mathbb{R}^D$ (*e.g.*, log-potentials for binary attributes), with $D \ll |\mathcal{Z}|$. Then, the score of some global configuration $z \in \mathcal{Z}$ is $s_z := \langle \boldsymbol{a}_z, \boldsymbol{t} \rangle$. The variable scores induce a unique Gibbs distribution over structures, given by $\pi(z|x, \theta) \propto \exp(\langle \boldsymbol{a}_z, \boldsymbol{t} \rangle)$. Equivalently, defining $\boldsymbol{A} \in \mathbb{R}^{D \times |\mathcal{Z}|}$ as the matrix with columns $\boldsymbol{a}_z$ for all $z \in \mathcal{Z}$, we consider the discrete distribution parameterized by $\mathsf{softmax}(\boldsymbol{s})$, where $\boldsymbol{s} = \boldsymbol{A}^\top \boldsymbol{t}$. (In the unstructured case, $\boldsymbol{A} = \boldsymbol{I}$.)

In practice, however, we cannot materialize the matrix $\boldsymbol{A}$ or the global score vector $\boldsymbol{s}$, let alone compute the softmax and the sum in Eq. 1. The SFE, however, can still be used, provided that exact sampling of $z \sim \pi(z|x, \theta)$ is feasible, and efficient algorithms exist for computing the normalizing constant $\sum_{z'} \exp(\langle \boldsymbol{a}_{z'}, \boldsymbol{t} \rangle)$ [24], needed to compute the probability of a given sample.

While it may be tempting to consider using sparsemax to avoid the expensive sum in the exact expectation, this is prohibitive too: solving the problem in Eq. 2 still requires explicit manipulation of the large vector $\boldsymbol{s} \in \mathbb{R}^{|\mathcal{Z}|}$, and even if we could avoid this, in the worst case ($\boldsymbol{s} = \boldsymbol{0}$) the resulting sparsemax distribution would still have exponentially large support. Fortunately, we show next that it is still possible to develop sparsification strategies to handle the combinatorial explosion of $\mathcal{Z}$ in the structured case. We propose two different methods to obtain a sparse distribution $\boldsymbol{\xi}$ supported only over a bounded-size subset of $\mathcal{Z}$: top-$k$ sparsemax (§4.1) and SparseMAP (§4.2).

## 4.1 Top-$k$ Sparsemax

Recall that the sparsemax operator (Eq. 2) is simply the Euclidean projection onto the $|\mathcal{Z}|$-dimensional probability simplex. While this projection has a propensity to be sparse, there is no upper bound on the number of non-zeros of the resulting distribution. When $\mathcal{Z}$ is large, one possibility is to add a cardinality constraint $\|\boldsymbol{\xi}\|_0 \leq k$ for some prescribed $k \in \mathbb{N}$. The resulting problem becomes

$$\mathsf{sparsemax}_k(\boldsymbol{s}) := \underset{\boldsymbol{\xi} \in \triangle^{|\mathcal{Z}|}, \|\boldsymbol{\xi}\|_0 \leq k}{\mathrm{argmin}} \|\boldsymbol{\xi} - \boldsymbol{s}\|_2^2, \tag{3}$$

which is known as a *sparse projection onto the simplex* and has been studied in detail by Kyrillidis et al. [25] and used to smooth structured prediction losses [26, 27]. Remarkably, while this is a non-convex problem, its solution $\boldsymbol{\xi}^\star$ can be written as a composition of two functions: a top-$k$ operator $\mathsf{top}_k : \mathbb{R}^{|\mathcal{Z}|} \to \mathbb{R}^{|\mathcal{Z}|}$, which returns a vector identical to its input but where all the entries not among the $k$ largest ones are masked out (set to $-\infty$), and the $k$-dimensional sparsemax operator.

Formally, $\mathsf{sparsemax}_k = \mathsf{sparsemax}(\mathsf{top}_k(\boldsymbol{s}))$. Being a composition of operators, its Jacobian becomes a product of matrices and hence simple to compute (the Jacobian of $\mathsf{top}_k$ is a diagonal matrix whose diagonal is a multi-hot vector indicating the top-$k$ elements of $\boldsymbol{s}$).

To apply the top-$k$ sparsemax to a large or combinatorial set $\mathcal{Z}$, all we need is a primitive to compute the top-$k$ entries of $\boldsymbol{s}$—this is available for many structured problems (for example, sequential models via $k$-best dynamic programming) and, when $\mathcal{Z}$ is the set of joint assignments of $D$ discrete binary variables, it can be done with a cost $\mathcal{O}(kD)$.

After enumerating this set, we parameterize $\pi(z|x, \theta)$ by applying sparsemax to that top-$k$, with a computational cost $\mathcal{O}(k)$. Note that **this method is identical to sparsemax whenever** $\|\mathsf{sparsemax}(\boldsymbol{s})\|_0 \leq k$: if during training the model learns to assign a sparse distribution to the latent variable, we are effectively using a sparsemax parameterization as presented in §3 with cheap computation. In fact, the solution of Eq. 3 gives us a certificate of optimality whenever $\|\boldsymbol{\xi}^\star\|_0 < k$.

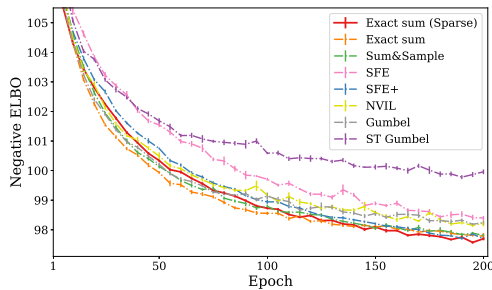

| Method | Accuracy (%) | Dec. calls |
|---|---|---|
| *Monte Carlo* | | |
| SFE | $94.75_{\pm.002}$ | 1 |
| SFE+ | $96.53_{\pm.001}$ | 2 |
| NVIL | $96.01_{\pm.002}$ | 1 |
| Sum&Sample | $96.73_{\pm.001}$ | 2 |
| Gumbel | $95.46_{\pm.001}$ | 1 |
| ST Gumbel | $86.35_{\pm.006}$ | 1 |
| *Marginalization* | | |
| Dense | $96.93_{\pm.001}$ | 10 |
| Sparse (proposed) | $96.87_{\pm.001}$ | $1.01_{\pm0.01}$ |

Figure 1: Semisupervised VAE on MNIST. Left: Learning curves (test). Right: Average test results and standard errors over 10 runs.

## 4.2 SparseMAP

A second possibility to obtain efficient summation over a combinatorial space without imposing any constraints on $\ell(x, z; \theta)$ is to use SparseMAP [13, 28], a structured extension of sparsemax:

$$\mathsf{SparseMAP}(\boldsymbol{t}) \coloneqq \operatorname*{argmin}_{\boldsymbol{\xi} \in \triangle^{|\mathcal{Z}|}} \|\boldsymbol{A}\boldsymbol{\xi} - \boldsymbol{t}\|_2^2, \quad (4)$$

SparseMAP has been used successfully in discriminative latent models to model structures such as trees and matchings, and Niculae et al. [13] apply an active set algorithm for evaluating it and computing gradients efficiently, requiring only a primitive for computing $\mathrm{argmax}_{z \in \mathcal{Z}} \langle \boldsymbol{a}_z, \boldsymbol{t} \rangle$. (We detail the algorithm in App. B). While the argmin in (4) is generally not unique, Carathéodory's theorem guarantees that solutions with support size at most $D + 1$ exist, and the active set algorithm enjoys linear and finite convergence to a very sparse optimal distribution. Crucially, (4) has a solution $\boldsymbol{\xi}^\star$ such that the set $\bar{\mathcal{Z}} = \{z \in \mathcal{Z} \mid \xi_z^\star > 0\}$ grows only linearly with $D$, and therefore $|\bar{\mathcal{Z}}| \ll |\mathcal{Z}|$. Therefore, assessing the expectation in Eq. 1 only requires evaluating $|\bar{\mathcal{Z}}| = \mathcal{O}(D)$ terms.

## 5 Experimental Analysis

We next demonstrate the applicability of our proposed strategies by tackling three tasks: a deep generative model with semisupervision (§5.1), an emergent communication two-player game over a discrete channel (§5.2), and a variational autoencoder with latent binary factors (§5.3). We describe any further architecture and hyperparameter details in App. E.

### 5.1 Semisupervised Variational Auto-encoder (VAE)

We consider the semisupervised VAE of Kingma et al. [29], which models the joint probability $p(z, h, x | \phi) = p(z)p(h)p(x | z, h)$, where $x$ is an observation (an MNIST image), $h$ is a continuous latent variable with a $n$-dimensional standard Gaussian prior, and $z$ is a discrete random variable with a uniform prior over $K$ categories. The marginal $p(x | \phi) = \sum_{z=1}^{K} \int_h p(x | z, h, \phi) p(h) p(z) \, \mathrm{d}h$ is intractable, due to the marginalization of $h \in \mathbb{R}^n$. For a fixed $h$ (*e.g.*, sampled), marginalizing $z$ requires $K$ calls to the decoder, which can be costly depending on the decoder's architecture.

To circumvent the need for the marginal likelihood, Kingma et al. [29] use variational inference [30] with an approximate posterior $\pi(z | x, \theta_\pi) q(h | z, x, \lambda)$. This trains a classifier $\pi(z | x, \theta_\pi)$ along with the generative model. In [29], $h$ is sampled with a reparameterization, and the expectation over $z$ is computed in closed-form, that is, assessing all $K$ terms of the sum for a sampled $h$. Under the notation in §2, we let $\theta_\ell = \{\lambda, \phi\}$ and define

$$\ell(x, z; \theta_\ell) \coloneqq -\mathbb{E}_{q(h|z,\lambda)} \left[\log p(x \mid z, h, \phi)\right] - \log \frac{p(z)}{\pi(z \mid x, \theta_\pi)} + \mathsf{KL}\left[q(h \mid x, z, \lambda) \parallel p(h)\right], \quad (5)$$

which turns Eq. 1 into the (negative) evidence lower bound (ELBO). To update $q(h | x, z, \lambda)$, we use the reparameterization trick to obtain gradients through a sampled $h$. For $\pi(z | x, \theta_\pi)$, we may still

explicitly marginalize over each possible assignment of $z$, but this has a multiplicative cost on $K$. As an alternative, we parameterize $\pi(z|x, \theta_\pi)$ with a sparse mapping, comparing it to the original formulation and with stochastic gradients based on SFE and continuous relaxations of $z$.

**Data and architecture.**   We evaluate this model on the MNIST dataset [31], using 10% of labeled data, treating the remaining data as unlabeled. For the parameterization of the model components we follow the architecture and training procedure used in [22]. Each model was trained for 200 epochs.

**Comparisons.**   Our proposal's key ingredient is sparsity, which permits exact marginalization and a deterministic gradient. To investigate the impact of sparsity alone, we report a comparison against the exact marginalization over the entire support $\mathcal{Z}$ using a dense softmax parameterization. To investigate the impact of deterministic gradients, we compare to stochastic gradients. Unbiased gradient estimators: *(i)* SFE with a moving average baseline; *(ii)* SFE with a self-critic baseline [SFE+; 32], that is, we use $\log p(x|z', h, \phi)$ as baseline, where $z' \sim \pi(z|x, \theta_\pi)$ is an independent sample; *(iii)* NVIL [33] with a learned baseline (we train a MLP to predict the learning signal by minimizing mean squared error); and *(iv)* sum-and-sample, a Rao-Blackwellized version of SFE [22]. Biased gradient estimators: *(v)* Gumbel-Softmax, which relaxes the random variable to the simplex, and *(vi)* ST Gumbel-Softmax, which discretizes the relaxation in the forward pass, but ignores the discretization function in the backward pass.[4]

**Results and discussion.**   In Fig. 1, we see that our proposed sparse marginalization approach performs just as well as its dense counterpart, both in terms of ELBO and accuracy. However, by inspecting the number of times each method calls the decoder for assessments of $p(x|z, h, \phi)$, we can see that the effective support of our method is much smaller — sparsemax-parameterized posteriors get very confident, and mostly require one, and sometimes two, calls to the decoder. Regarding the Monte Carlo methods, the continuous relaxation done by Gumbel-Softmax underperformed all the other methods, with the exception of SFE with a moving average. While SFE+ and Sum&Sample are very strong performers, they will always require throughout training the same number of calls to the decoder (in this case, two). On the other hand, sparsemax makes a small number of decoder calls not due to a choice in hyperparameters but thanks to the model converging to only using a small support, which can endow this method with a lower number of computations as it becomes more confident.

## 5.2   Emergent Communication Game

Emergent communication studies how two agents can develop a communication protocol to solve a task collaboratively [34]. Recent work used neural latent variable models to train these agents via a "collaborative game" between them [35–40]. In [36], one of the agents (the *sender*) sees an image $v_y$ and sends a single symbol message $z$ chosen from a set $\mathcal{Z}$ (the *vocabulary*) to the other agent (the *receiver*), who needs to choose the correct image $v_y$ out of a collection of images $\mathcal{V} = \{v_1, \ldots, v_C\}$.[5] They found that the messages communicated this way can be correlated with broad object properties amenable to interpretation by humans. In our framework, we let $x = (\mathcal{V}, y)$ and define $\ell(x, z; \theta) := -\log p(y \mid \mathcal{V}, z, \theta_\ell)$ and $\pi(z \mid x, \theta) := p(z \mid v_y, \theta_\pi)$, where $p(y \mid \mathcal{V}, z, \theta_\ell)$ corresponds to the sender and $p(z \mid v_y, \theta_\pi)$ to the receiver. Following Lazaridou et al. [36], we add an entropy regularization of $\pi(z \mid x, \theta)$ to the loss, with a coefficient as an hyperparameter [41].

**Data and architecture.**   We follow the architecture described in [36]. However, to make the game harder, we increase the collection of images $|\mathcal{V}|$ as suggested by [37]; in our experiments, we increase it from 2 to 16. All methods are trained for 500 epochs.

**Comparisons.**   We compare our method to stochastic gradient estimators as well as exact marginalization under a dense softmax parameterization of $p(z \mid v_y, \theta_\pi)$. Again, we have unbiased (SFE with

moving average baseline, SFE+, and NVIL) and biased (Gumbel-Softmax and ST Gumbel-Softmax) estimators. For SFE we also experiment with a 0/1 loss, rather than negative log-likelihood (NLL).

**Results and discussion.** Table 1 shows the communication success (accuracy of the receiver at picking the correct image $v_y$). While the communication success for $|\mathcal{V}| = 2$ in [36] was close to perfect, we see that increasing $|\mathcal{V}|$ to 16 makes this game much harder to sampling-based approaches. In fact, only the models that do explicit marginalization achieve close to perfect communication in the test set. However, as $\mathcal{Z}$ increases, marginalizing with a softmax parameterization gets computationally more expensive, as it requires $|\mathcal{Z}|$ forward and backward passes on the receiver. Unlike softmax, the model trained with sparsemax gives a very small support, requiring on

Table 1: Emergent communication success test results, averaged across 10 runs. Random guess baseline 6.25%.

| Method | Comm. succ. (%) | Dec. calls |
|---|---|---|
| *Monte Carlo* | | |
| SFE (NLL) | $33.05_{\pm 2.84}$ | 1 |
| SFE (0/1) | $55.36_{\pm 2.92}$ | 1 |
| SFE+ (0/1) | $44.32_{\pm 2.72}$ | 2 |
| NVIL | $37.04_{\pm 1.61}$ | 1 |
| Gumbel | $23.51_{\pm 16.19}$ | 1 |
| ST Gumbel | $27.42_{\pm 13.36}$ | 1 |
| *Marginalization* | | |
| Dense | $93.37_{\pm 0.42}$ | 256 |
| Sparse (proposed) | $93.35_{\pm 0.50}$ | $3.13_{\pm 0.48}$ |

average only 3 decoder calls. In fact, sparsemax starts off dense while exploring, but quickly becomes very sparse (App. F).

## 5.3 Bit-Vector Variational Autoencoder

As described in §4, in many interesting problems, combinatorial interactions and constraints make $\mathcal{Z}$ exponentially large. In this section, we study the illustrative case of encoding (compressing) images into a binary codeword $z$, by training a latent bit-vector variational autoencoder [11, 33]. One approach for parameterizing the approximate posterior is to use a Gibbs distribution, decomposable as a product of independent Bernoulli, $q(z \mid x, \lambda) \propto \exp(\langle \boldsymbol{a}_z, \boldsymbol{t} \rangle) = \prod_{i=1}^{D} q(z_i \mid x, \lambda)$, with each $z_i$ being a Bernoulli with parameter $t_i$, and $D$ being the number of binary latent variables. While marginalizing over all the possible $z \in \mathcal{Z}$ is intractable, drawing samples can be done efficiently by sampling each component independently, and the entropy has a closed-form expression. This efficient sampling and entropy computation relies on an independence assumption; in general, we may not have access to such efficient computation.

Training this VAE to minimize the negative ELBO corresponds to $\ell(x, z; \theta_\ell) := -\log \frac{p(x, z|\phi)}{q(z|x, \lambda)}$; we use a uniform prior $p(z) = 1/|\mathcal{Z}| = 1/2^D$. This objective does not constrain $\pi(z|x, \theta_\pi) := q(z \mid x, \lambda)$ to the Gibbs parameterization, and thus to apply our methods we will differ from it.

**Top-$k$ sparsemax parameterization.** As pointed out in §4, we cannot explicitly handle the structured sparsemax distribution $\boldsymbol{\xi} = \mathsf{sparsemax}(\boldsymbol{s})$, as it involves a vector of dimension $2^D$. However, given $\boldsymbol{t}$, we can efficiently find the $k$ largest configurations in time $\mathcal{O}(kD)$, with the procedure described in §4.1, and thus we can evaluate $\mathsf{sparsemax}_k(\boldsymbol{s})$ efficiently.

**SparseMAP parameterization.** Another sparse alternative to the intractable structured sparsemax, as discussed in §4, is SparseMAP. In this case, we compute an optimal distribution $\boldsymbol{\xi}$ using the active set algorithm of [13], by using a maximization oracle which can be computed in $\mathcal{O}(D)$:

$$\operatorname*{argmax}_z \langle \boldsymbol{a}_z, \boldsymbol{t} \rangle = z^\star \quad \text{s.t.} \quad [\boldsymbol{a}_{z^\star}]_i = \begin{cases} 1, & t_i \geq 0 \\ 0, & t_i < 0 \end{cases}. \tag{6}$$

Since SparseMAP can handle structured problems, we also experimented with adding a *budget constraint* to SparseMAP: this is done by adding a constraint $\|\boldsymbol{z}\|_1 \leq B$, where $B \leq D$; we used $b = \frac{D}{2}$. The budget constraint ensures the images are represented with sparse codes, and the maximization oracle can be computed in $\mathcal{O}(D \log D)$ as described in App. C.

We stress that, with both top-$k$ sparsemax and SparseMAP parameterizations, $z$ does not decompose into a product of independent binary variables: this property is specific to the Gibbs parameterization.

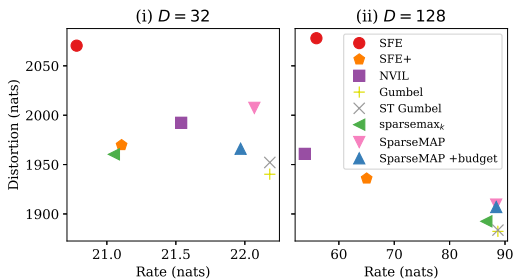

| Method | $D = 32$ | $D = 128$ |
|---|---|---|
| *Monte Carlo* | | |
| SFE | 3.74 | 3.77 |
| SFE+ | 3.61 | 3.59 |
| NVIL | 3.65 | 3.60 |
| Gumbel | 3.57 | 3.49 |
| ST Gumbel | 3.53 | 3.55 |
| *Marginalization* | | |
| Top-$k$ sparsemax | 3.62 | 3.61 |
| SparseMAP | 3.72 | 3.67 |
| SparseMAP (w/ budget) | 3.64 | 3.66 |

Figure 2: Test results for Fashion-MNIST. Left and middle: RD plots (the closer to the lower right corner, the better). Right: NLL in bits/dim (lower, the better).

However, since these new approaches induce a very sparse approximate posterior $q$, we may compute the terms $\mathbb{E}_{q(z|x,\lambda)}[\log p(x \mid z, \phi)]$ and $\mathbb{E}_{q(z|x,\lambda)}[\log q(z \mid x, \lambda)]$ explicitly.

**Data and architecture.** We use Fashion-MNIST [42], consisting of 256-level grayscale images $x \in \{0, 1, \dots, 255\}^{28 \times 28}$. The decoder uses an independent categorical distribution for each pixel, $p(x \mid z, \phi) = \prod_{i=1}^{28} \prod_{j=1}^{28} p(x_{ij} \mid z, \phi)$. For top-$k$ sparsemax, we choose $k = 10$.

**Comparisons.** This time, exact marginalization under a dense parameterization of $q(z|x, \lambda)$ is truly intractable, so we can only compare our method to stochastic gradient estimators. We have unbiased SFE-based estimators (SFE with moving average baseline, SFE+, and NVIL), and biased reparameterized gradient estimators (Gumbel-Softmax and ST Gumbel-Softmax). As there is no supervision for the latent code, we cannot compare the methods in terms of accuracy or task success. Instead, we display the trained models in the rate-distortion (RD) plane [43][6] and also report bits-per-dimension of $x$, estimated with importance sampling (App. D), on held-out data.

**Results and discussion.** Fig. 2 shows an importance sampling estimate (1024 samples per test example were taken) of the negative log-likelihood for the several methods, together with the converged values of each method in the RD plane. Both show results for which the bit-vector has dimensionality $D = 32$ and $D = 128$. Regarding the estimated negative log-likelihood, our methods exhibit increased performance when compared to SFE, and top-$k$ sparsemax is competitive with the remaining unbiased estimators. However, in the RD plane, both our methods show comparable performance to SFE+ and NVIL for $D = 32$, but for $D = 128$ all of our methods have a significantly higher rate and lower distortion than any unbiased estimator, suggesting a better fit of $p(x|\phi)$ [43]. In Fig. 3, we can observe the training

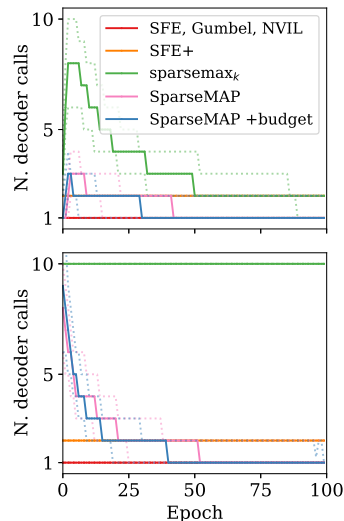

Figure 3: Bit vector VAE median and quartile decoder calls per epoch, $D = 32$ (top) / $D = 128$ (bottom).

progress in number of calls to $p(x \mid z, \phi)$ for the models with 32 and 128 latent bits, respectively. While sparsemax$_k$ introduces bias towards the most probable assignments and may discard outcomes that sparsemax would assign non-zero probability to, as training progresses distributions may (or tend to) be sufficiently sparse and this mismatch disappears, making the gradient computation exact. Remarkably, this happens for $D = 32$ — the support of sparsemax$_k$ is smaller than $k$, giving the true gradient to $q(z \mid x, \lambda)$ for most of the training. This no longer happens for $D = 128$, for which it remains with full support throughout, due to the much larger search space. On the other

hand, SparseMAP solutions become very sparse from the start in both cases, while still obtaining good performance. There is, therefore, a trade-off between the solutions we propose: on one hand, $\text{sparsemax}_k$ can become exact with respect to the expectation in Eq. 1, but it only does so if the chosen $k$ is suitable to the difficulty of the model; on the other hand, SparseMAP may not offer an exact gradient to $q(z \mid x, \lambda)$, but its performance is very close to $\text{sparsemax}_k$ and its higher propensity for sparsity gifts it with less computation (App. F).

Concerning relaxed estimators, note that the reconstruction loss is computed given a continuous sample, rather than a discrete one, allowing it more flexibility to directly reduce distortion and potentially explaining why it does well in that regard. Moreover, the rate of the relaxed model is unknown,[7] and instead we plot the rate as if $z$ was given discrete treatment, which, although common practice, makes comparisons to other estimators inadequate. For ST Gumbel-Softmax the situation is different since, after training, $z$ is given discrete treatment throughout. Its success shows that, unlike in the other tasks considered, training on biased gradients is not too problematic.

## 6  Related Work

**Differentiable sparse mappings.**   There has been recent interest in applying sparse mappings of discrete distributions in deep discriminative models [12, 13, 44, 45, 28], attention mechanisms [46–49], and in topic models [50]. Our work focuses on the parameterization of distributions over latent variables with sparse mappings, on the computational advantage to be gained by sparsity, and on the contrast between our novel training method and common sampling-based methods.

**Reducing sampling noise.**   The sampling procedure found in SFE is a great source of variance in models that build upon this method. To reduce this variance, many works have proposed baselines [15–17]. VIMCO [51] is a multi-sample estimator which exploits variance reduction via input-dependent baselines as well as a lower bound on marginal likelihood which is tighter than the ELBO [52]. The number of samples in VIMCO is a hyperparameter that stays fixed throughout training. Our methods, in contrast, may take several decoder calls initially, but that number automatically decreases over time as training progresses. While baselines must be independent of the sample for which we assess the score function, exploiting correlation in the downstream losses of dependent samples holds potential for further variance reduction. These are known as control variates [53]. REBAR [18] exploits a continuous relaxation to obtain a dependent sample and uses the downstream loss assessed at the relaxed sample to define a control variate. RELAX [19], instead, learns to predict the downstream loss of the relaxed sample with an auxiliary network. In contrast, sparse marginalization works for any factorization where a primitive for 1-best (or $k$-best) enumeration is available, and takes no additional parameters nor additional optimization objectives. Another line of work approximates argmax gradients by perturbed finite differences [54, 55]; this requires the same computation primitive as our approach, but is always biased. ARM [56] is a control variate based on antithetic samples [57]: it does not require relaxation nor additional parameters, but it only applies to factorial Bernoulli distributions. Closest to our work are variance reduction techniques that rely on partial marginalization, typically of the top-$k$ assignments to the latent variable [22, 58]. These methods show improved performance and variance reduction, but require rejection sampling, which can be challenging in structured problems.

## 7  Conclusion

We described a novel training strategy for discrete latent variable models, eschewing the common approach based on MC gradient estimation in favor of deterministic, exact marginalization under a sparse distribution. Sparsity leads to a powerful *adaptive* method, which can investigate fewer or more latent assignments $z$ depending on the ambiguity of a training instance $x$, as well as on the stage in training. We showcase the performance and flexibility of our method by investigating a variety of applications, with both discrete and structured latent variables, with positive results. Our models very quickly approach a small number of latent assignment evaluations per sample, but make progress much faster and overall lead to superior results. Our proposed method thus offer the accuracy and robustness of exact marginalization while meeting the efficiency and flexibility of score function estimator methods, providing a promising alternative.

## Broader Impact

We discuss here the broader impact of our work. Discussion in this section is predominantly speculative, as the methods described in this work are not yet tested in broader applications. However, we do think that the methods described here can be applied to many applications — as this work is applicable to any model that contains discrete latent variables, even of combinatorial type.

Currently, the solutions available to train discrete latent variable models (LVMs) greatly rely on MC sampling, which can have high variance. Methods that aim to decrease this variance are often not trivial to train and to implement and may disincentivize practitioners from using this class of models. However, we believe that discrete LVMs have, in many cases, more interpretable and intuitive latent representations. Our methods offer: a simple approach in implementation to train these models; no addition in the number of parameters; low increase in computational overhead (especially when compared to more sophisticated methods of variance reduction [22]); and improved performance. Our code has been open-sourced as to ensure it's scrutinizable by anyone and to boost any related future work that other researchers might want to pursue.

As we have already pointed out, oftentimes LVMs have superior explanatory power and so can aid in understanding cases in which the model failed the downstream task. Interpretability of deep neural models can be essential to better discover any ethically harmful biases that exist in the data or in the model itself.

On the other hand, the generative models discussed in this work may also pave the way for malicious use cases, such as is the case with *Deepfakes*, fake human avatars used by malevolent Twitter users, and automatically generated fraudulent news. Generative models are remarkable at sampling new instances of fake data and, with the power of latent variables, the interpretability discussed before can be used maliciously to further push harmful biases instead of removing them. Furthermore, our work is promising in improving the performance of LVMs with several discrete variables, that can be trained as attributes to control the sample generation. Attributes that can be activated or deactivated at will to generate fake data can both help beneficial and malignant users to finely control the generated sample. Our work may be currently agnostic to this, but we recognize the dangers and dedicate effort to combating any malicious applications.

Energy-wise, LVMs often require less data and computation than other models that rely on a massive amount of data and infrastructure. This makes LVMs ideal for situations where data is scarce, or where there are few computational resources to train large models. We believe that better latent variable modeling is a step forward in the direction of alleviating environmental concerns of deep learning research [59]. However, the models proposed in this work tend to use more resources earlier on in training than standard methods, and even though in the applications shown they consume much less as training progresses, it's not clear if that trend is still observed in all potential applications.

In data science, LVMs, such as mixed-membership models [60], can be used to uncover correlations in large amounts of data, for example, by clustering observations. Training these models requires various degrees of approximations which are not without consequence, they may impact the quality of our conclusions and their fealty to the data. For example, variational inference tends to under-estimate uncertainty and give very little support to certain less-represented groups of variables. Where such a model informs decision-makers on matters that affect lives, these decisions may be based on an incomplete view of the correlations in the data and/or these correlations may be exaggerated in harmful ways. On the one hand, our work contributes to more stable training of LVMs, and thus it is a step towards addressing some of the many approximations that can blur the picture. On the other hand, sparsity may exhibit a larger risk of disregarding certain correlations or groups of observations, and thus contribute to misinforming the data scientist. At this point it is unclear to which extent the latter happens and, if it does, whether it is consistent across LVMs and their various uses. We aim to study this issue further and work with practitioners to identify failure cases.

## Acknowledgments and Disclosure of Funding

Top-$k$ sparsemax is due in great part to initial work and ideas of Mathieu Blondel. The authors are thankful to Wouter Kool for feedback and suggestions. We are grateful to Ben Peters, Erick Fonseca, Marcos Treviso, Pedro Martins, and Tsvetomila Mihaylova for insightful group discussion. We would also like to thank the anonymous reviewers for their helpful feedback.

This work was partly funded by the European Research Council (ERC StG DeepSPIN 758969), by the P2020 project MAIA (contract 045909), and by the Fundação para a Ciência e Tecnologia through contract UIDB/50008/2020. This work also received funding from the European Union's Horizon 2020 research and innovation programme under grant agreement 825299 (GoURMET).

## Footnotes

*Work partially done while VN was at the Instituto de Telecomunicações, Lisbon.

[2]Code is publicly available at `https://github.com/deep-spin/sparse-marginalization-lvm`

[3]Not to be confused with "score function," as in SFE, which refers to the gradient of the log-likelihood.

[4] For Gumbel-Softmax (with and without ST), we follow Jang et al. [11] and substitute $\mathsf{KL}(\pi(z|x, \theta_\pi)\|p(z))$ in the ELBO by the KL divergence of $\mathsf{Categorical}(\mathsf{softmax}(s))$ from a discrete uniform prior. Strictly speaking this means the objective is not a proper ELBO and its relationship to an ELBO is unclear [10, Appendix C.2].

[5] Lazaridou et al. [36] lets the sender see the full set $\mathcal{V}$. In contrast, we follow [37] in showing only the correct image $v_y$ to the sender. This makes the game harder, as the message $z$ needs to encode a good "description" of the correct image $v_y$ instead of encoding only its differences from $\mathcal{V} \setminus \{v_y\}$.

[6]Distortion is the expected value of the reconstruction negative log-likelihood, while rate is the average KL divergence from the prior to the approximate posterior.

[7]Estimating it would require a choice of Binary Concrete prior and an estimate of the KL divergence from that to the Binary Concrete approximate posterior [10, Appendix C.3.2].

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
