[Supplementary Material]

# A    Computing sparsemax: Forward and Backward Passes

The sparsemax mapping [12], as discussed in Section 3, is given by the unique solution to

$$\text{sparsemax}(\boldsymbol{s}) \coloneqq \underset{\boldsymbol{\xi} \in \triangle^K}{\arg\min} \|\boldsymbol{\xi} - \boldsymbol{s}\|_2^2 \,. \tag{7}$$

As a projection onto a polytope, the solution is likely to fall on the boundaries or the corners of the set. In this case, points on the boundary of $\triangle^K$ have one or more zero coordinates. In contrast, $\text{softmax}(\boldsymbol{s}) \propto \exp(\boldsymbol{s})$ is always strictly inside the simplex. From the optimality conditions of the sparsemax problem (7), it follows that the solution must have the form:

$$\text{sparsemax}(\boldsymbol{s}) = \max(\boldsymbol{s} - \tau, 0) \,, \tag{8}$$

where the maximum is elementwise, and $\tau$ is the unique value that ensures the result sums to one. Letting $\bar{\mathcal{Z}}$ be the set of nonzero coordinates in the solution, the normalization condition is equivalently

$$\tau = \frac{\sum_{z \in \bar{\mathcal{Z}}} s_z}{|\bar{\mathcal{Z}}|} \,. \tag{9}$$

Observing that small changes to $\boldsymbol{s}$ almost always have no effect on the support $\bar{\mathcal{Z}}$, differentiating Equation 8 gives

$$\frac{\partial \bar{\boldsymbol{\xi}}}{\partial \bar{\boldsymbol{s}}} = \boldsymbol{I}_{|\bar{z}|} - \frac{1}{|\bar{\mathcal{Z}}|} \boldsymbol{1}\boldsymbol{1}^\top \,, \tag{10}$$

where $\bar{\boldsymbol{\xi}}$ and $\bar{\boldsymbol{s}}$ denote the subsets of the respective vectors indexed by the support $\bar{\mathcal{Z}}$. Outside of the support, the partial derivatives are zero. (*Cf.* the more general result in [45, Proposition 2].) In terms of computation, $\tau$ may be found numerically using root finding algorithms on $f(\tau) = \max(\boldsymbol{s} - \tau, 0) - 1$. Alternatively, observe that it is enough to find $\bar{\mathcal{Z}}$. By showing that sparsemax must preserve the ordering, *i.e.*, that if $s_{z'} > s_z$ and $z \in \bar{\mathcal{Z}}$ then $z' \in \bar{\mathcal{Z}}$, it can be shown that $\bar{\mathcal{Z}}$ must consist of the $k$ highest-scoring coordinates of $\boldsymbol{s}$, where $k$ can be find by inspection after sorting $\boldsymbol{s}$. This leads to a straightforward $\mathcal{O}(K \log K)$ algorithm due to Held [23, pp. 16–17]. This can be further pushed to $\mathcal{O}(K)$ using median pivoting algorithms [61]. We use a simpler implementation based on repeatedly calling $\text{top}_k$, doubling $k$ until the optimal solution is found. Since solutions get sparser over time and $\text{top}_k$ is GPU-accelerated in modern libraries [62], this strategy is very fast in practice.

# B    Computing SparseMAP: The Active Set Algorithm

In this section, we present the active set method [63, Chapters 16.4 & 16.5] as applied to the SparseMAP optimization problem (Eq. 4) [13]. This form of the algorithm, due to Martins et al. [64, Section 6], is a small variation of the formulation of Nocedal and Wright for handling the equality constraint. Recall the SparseMAP problem,

$$\underset{\boldsymbol{\xi} \in \triangle^{|\mathcal{Z}|}}{\arg\min} \|\boldsymbol{A}\boldsymbol{\xi} - \boldsymbol{t}\|_2^2 \,. \tag{11}$$

Assume that we could identify the *support*, or *active set* of an optimal solution $\boldsymbol{\xi}^\star$, denoted

$$\bar{\mathcal{Z}} \coloneqq \{z \in \mathcal{Z} \mid \xi_z^\star > 0\} \,.$$

Then, given this set, we could find the solution to (11) by solving the lower-dimensional equality-constrained problem

$$\text{minimize} \|\bar{\boldsymbol{A}}\bar{\boldsymbol{\xi}} - \boldsymbol{t}\|^2 \quad \text{s.t.} \quad \boldsymbol{1}^\top \bar{\boldsymbol{\xi}} = 1 \,, \tag{12}$$

where we denote by $\bar{\boldsymbol{A}}$ and $\bar{\boldsymbol{\xi}}$ the restrictions of $\boldsymbol{A}$ and $\boldsymbol{\xi}$ to the active set of structures $\bar{\mathcal{Z}}$. The solution to this equality-constrained QP satisfies the KKT optimality conditions,

$$\begin{bmatrix} \bar{\boldsymbol{A}}^\top \bar{\boldsymbol{A}} & \boldsymbol{1} \\ \boldsymbol{1}^\top & 0 \end{bmatrix} \begin{bmatrix} \bar{\boldsymbol{\xi}} \\ \tau \end{bmatrix} = \begin{bmatrix} \bar{\boldsymbol{A}}^\top \boldsymbol{t} \\ 1 \end{bmatrix} \,. \tag{13}$$

Of course, the optimal support is not known ahead of time. The active set algorithm attempts to guess the support in a greedy fashion, at each iteration either [if the solution of (13) is not feasible for (11)] dropping a structure from $\bar{\mathcal{Z}}$, or [otherwise] adding a new structure. Since the support changes one structure at a time, the design matrix in (13) gains or loses one row and column, so we may efficiently maintain its Cholesky decomposition via rank-one updates.

We now give more details about the computation. Denote the solution of Eq. 13, (extended with zeroes), by $\hat{\boldsymbol{\xi}} \in \triangle^{|\mathcal{Z}|}$. Since we might not have the optimal $\bar{\mathcal{Z}}$ yet, $\hat{\boldsymbol{\xi}}$ can be infeasible (some coordinates may be negative.) To account for this, we take a partial step in its direction,

$$\boldsymbol{\xi}^{(i+1)} = (1 - \gamma)\boldsymbol{\xi}^{(i)} + \gamma\hat{\boldsymbol{\xi}}^{(i+1)} \tag{14}$$

where, to ensure feasibility, the step size is given by

$$\gamma = \min\left(1, \min_{z\in\bar{\mathcal{Z}};\xi_z^{(i)}>\hat{\xi}_z} \frac{\xi_z^{(i)}}{\xi_z^{(i)} - \hat{\xi}_z}\right). \tag{15}$$

If, on the other hand, $\hat{\boldsymbol{\xi}}$ is feasible for (11), (so $\gamma = 1$), we check whether we have a globally optimal solution. By construction, $\hat{\boldsymbol{\xi}}$ satisfies all KKT conditions except perhaps dual feasibility $\boldsymbol{\nu} \geq 0$, where $\nu_z$ is the dual variable (Lagrange multiplier) corresponding to the constraint $\xi_z \geq 0$. Denote $\boldsymbol{\mu}^{(i)} := \boldsymbol{A}\boldsymbol{\xi}^{(i)} = \bar{\boldsymbol{A}}\bar{\boldsymbol{\xi}}^{(i)}$. For any $z \notin \bar{\mathcal{Z}}$, the corresponding dual variable must satisfy

$$\nu_z = \tau^{(i)} - \langle \boldsymbol{a}_z, \boldsymbol{t} - \boldsymbol{\mu}^{(i)} \rangle. \tag{16}$$

If the smallest dual variable is positive, then our current guess satisfies all optimality conditions. To find the smallest dual variable we can equivalently solve $\mathrm{argmax}_{z\in\mathcal{Z}}\langle \boldsymbol{a}_z, \boldsymbol{t} - \boldsymbol{\mu}^{(i)} \rangle$, which is a maximization (MAP) oracle call. If the resulting $\nu_z$ is negative, then $z$ is the index of the most violated constraint $\xi_z \geq 0$; it is thus a good choice of structure to add to the active set.

The full procedure is given in Algorithm 1. The backward pass can be computed by implicit differentiation of the KKT system (13) *w.r.t.* $\boldsymbol{t}$, giving, as in [13],

$$\frac{\partial\bar{\boldsymbol{\xi}}}{\partial\boldsymbol{t}} = \bar{\boldsymbol{A}}\big(\boldsymbol{S} - \boldsymbol{ss}^\top/s\big), \quad \text{where} \quad \boldsymbol{S} = (\bar{\boldsymbol{A}}^\top\bar{\boldsymbol{A}})^{-1}, \boldsymbol{s} = \boldsymbol{S}\mathbf{1}, s = \mathbf{1}^\top\boldsymbol{S}\mathbf{1}. \tag{17}$$

It is possible to apply the $\ell_2$ regularization term only to a subset of the rows of $\boldsymbol{A}$, as is more standard in the graphical model literature. We refer the reader to the presentation in [64, 13] for this extension.

---

**Algorithm 1** Active set algorithm for SparseMAP

    **Init:** $\bar{\mathcal{Z}}^{(0)} = \{z^{(0)}\}$    where    $z^{(0)} \in \mathrm{argmax}_{z\in\mathcal{Z}}\langle \boldsymbol{a}_z, \boldsymbol{t} \rangle$ or a random structure.
1:  **for** i in $1, \ldots, N$ **do**
2:      Compute $\tau^{(i)}$ and $\hat{\boldsymbol{\xi}}^{(i)}$ by solving the relaxed QP (Eq. 13).                 $\triangleright$ Cholesky update.
3:      $\boldsymbol{\xi}^{(i)} \leftarrow (1 - \gamma)\boldsymbol{\xi}^{(i-1)} + \gamma\hat{\boldsymbol{\xi}}^{(i)}$  (with $\gamma$ from Eq. 15).
4:      **if** $\gamma < 1$ **then**
5:          Drop the minimizer of Eq. 15 from $\bar{\mathcal{Z}}^{(i)}$.
6:      **else**
7:          Find most violated constraint, $z^{(i)} \leftarrow \mathrm{argmin}_{z\in\mathcal{Z}}\nu_z$.          $\triangleright$ Eq. 16, MAP oracle.
8:          **if** $\nu_{z^{(i)}} \geq 0$ **then**
9:              **return**                                                $\triangleright$ Converged.
10:         **else**
11:            $\mathcal{Z}^{(i+1)} \leftarrow \mathcal{Z}^{(i)} \cup \{z^{(i)}\}$

---

## C    Budget Constraint

The maximization oracle for the budget constraint described in §5.3 can be computed in $\mathcal{O}(D \log D)$. This is done by sorting the Bernoulli scores and selecting the entries among the top-$B$ which have a positive score.

## D    Importance Sampling of the Marginal Log-Likelihood

Bits-per-dimension is the negative logarithm of marginal likelihood normalized per number of pixels in the image, thus we need to assess or estimate the marginal likelihood of observations. For dense parameterizations, the usual option is importance sampling (IS) using the trained approximate posterior as importance distribution: *i.e.*, $\log p(x|\phi) \overset{\text{IS}}{\approx} \log\left(\frac{1}{S}\sum_{s=1}^{S}\frac{p(z^{(s)}, x|\phi)}{q(z^{(s)}|x,\lambda)}\right)$ with $z^{(s)} \sim q(z|x, \lambda)$. The

result is a stochastic lower bound which converges to the true log-marginal in the limit as $S \to \infty$. With a sparse posterior approximation we can split the marginalization

$$\log p(x|\phi) = \log \left( \sum_{z \in \bar{\mathcal{Z}}} p(z)p(x|z,\phi) + \sum_{z \in \mathcal{Z} \setminus \bar{\mathcal{Z}}} p(z)p(x|z,\phi) \right) \tag{18}$$

into one part that handles outcomes in the support $\bar{\mathcal{Z}}$ of the sparse posterior approximation and another part that handles the outcomes in the complement set $\mathcal{Z} \setminus \bar{\mathcal{Z}}$. We compute the first part exactly and estimate the second part via rejection sampling from $p(z)$.

# E Training Details

In our applications, we follow the experimental procedures described in [22] and [36] for §5.1 and §5.2, respectively. We describe below the most relevant training details and key differences in architectures when applicable. For other implementation details that we do not mention here, we refer the reader to the works referenced above. For all Gumbel baselines, we relax the sample into the continuous space but assume a discrete distribution when computing the entropy of $\pi(z \mid x, \theta)$, as suggested as one implementation option in Maddison et al. [10]. Our code is publicly available at `https://github.com/deep-spin/sparse-marginalization-lvm` and was largely inspired by the structure and implementations found in EGG [65] and was built upon it.

**Semisupervised Variational Autoencoder.** In this experiment, the classification network consists of three fully connected hidden layers of size 256, using ReLU activations. The generative and inference network both consist of one hidden layer of size 128, also with ReLU activations. The multivariate Gaussian has 8 dimensions and its covariance is diagonal. For all models we have chosen the learning rate based on the best ELBO on the validation set, doing a grid search (5e-5, 1e-4, 5e-4, 1e-3, 5e-3). The accuracy shown in Fig. 1 is the test accuracy taken after the last epoch of training. The temperature of the Gumbel models was annealed according to $\tau = \max(0.5, -rt)$, where $t$ is the global training step. For these models, we also did a grid search over $r$ (1e-5, 1e-4) and over the frequency of updating $\tau$ every (500, 1000) steps. Optimization was done with Adam. For our method, in the labeled loss component of the semisupervised objective we used the sparsemax loss [12]. Following Liu et al. [22], we pretrain the network with only labeled data prior to training with the whole training set. Likewise, for our method, we pretrained the network on the sparsemax loss and every other method with the Negative Log-Likelihood loss.

**Emergent communication game.** In this application, we closely followed the experimental procedure described by Lazaridou et al. [36] with a few key differences. The architecture of the sender and the receiver is identical with the exception that the sender does not take as input the distracting images along with the correct image — only the correct image. The collection of images shown to the receiver was increased from 2 to 16 and the vocabulary of the sender was increased to 256. The hidden size and embedding size was also increased to 512 and 256, respectively. We did a grid search on the learning rate (0.01, 0.005, 0.001) and entropy regularizer (0.1, 0.05, 0.01) and chose the best configuration for each model on the validation set based on the communication success. For the Gumbel models, we applied the same schedule and grid search to the temperature as described for Semisupervised VAE. All models were trained with the Adam optimizer, with a batch size of 64 and during 200 epochs. We choose the vocabulary of the sender to be 256, the hidden size to be 512 and the embedding size to be 256.

**Bit-Vector Variational Autoencoder.** In this experiment, we have set the generative and inference network to consist of one hidden layer with 128 nodes, using ReLU activations. We have searched a learning rate doing grid search (0.0005, 0.001, 0.002) and chosen the model based on the ELBO performance on the validation set. For the Gumbel models, we applied the same schedule and grid search to the temperature as described for Semisupervised VAE. We used the Adam optimizer.

## E.1 Datasets

**Semisupervised Variational Autoencoder.** MNIST consists of $28 \times 28$ gray-scale images of hand-written digits. It contains 60,000 datapoints for training and 10,000 datapoints for testing. We perform model selection on the last 10,000 datapoints of the training split.

**Emergent communication game.** The data used by Lazaridou et al. [36] is a subset of ImageNet containing 463,000 images, chosen by sampling 100 images from 463 base-level concepts. The images are then applied a forward-pass through the pretrained VGG ConvNet [66] and the representations at the second-to-last fully connected layer are saved to use as input to the sender/receiver.

**Bit-Vector Variational Autoencoder.** Fashion-MNIST consists of $28 \times 28$ gray-scale images of clothes. It contains 60,000 datapoints for training and 10,000 datapoints for testing. We perform model selection on the last 10,000 datapoints of the training split.

# F    Performance in Decoder Calls

Fig. 4 shows the number of decoder calls with percentiles for the experiment in §5.2. While dense right at the beginning of training, support quickly falls to an average of close to 1 decoder call.

Fig. 5 shows the downstream loss (ELBO) of experiment §5.3 over epochs and over the median number of decoder calls per epoch. The plots on Fig. 5b show how our methods have comparable computational overhead to sampling approaches. Oftentimes, our methods could have been trained in less epochs to obtain the same performance as the sampling estimators have for 100 epochs.

Figure 4: Median decoder calls per epoch during training time with 10 and 90 percentiles in dotted lines by sparsemax in §5.2.

# G    Computing infrastructure

Our infrastructure consists of 4 machines with the specifications shown in Table 2. The machines were used interchangeably, and all experiments were executed in a single GPU. Despite having machines with different specifications, we did not observe large differences in the execution time of our models across different machines.

| #  | GPU                    | CPU                                         |
|----|------------------------|---------------------------------------------|
| 1. | $4 \times$ Titan Xp - 12GB      | $16 \times$ AMD Ryzen 1950X @ 3.40GHz - 128GB       |
| 2. | $4 \times$ GTX 1080 Ti - 12GB   | $8 \times$ Intel i7-9800X @ 3.80GHz - 128GB         |
| 3. | $3 \times$ RTX 2080 Ti - 12GB   | $12 \times$ AMD Ryzen 2920X @ 3.50GHz - 128GB       |
| 4. | $3 \times$ RTX 2080 Ti - 12GB   | $12 \times$ AMD Ryzen 2920X @ 3.50GHz - 128GB       |

Table 2: Computing infrastructure.

(a) Negative ELBO over training epochs.

(b) Negative ELBO over decoder calls.

Figure 5: Performance on the validation set for the experiment in §5.3, $D = 32$ (top) / $D = 128$ (bottom). For $D = 32$, top-$k$ sparsemax continues until a total of 561 median decoder calls, and for $D = 128$ it continues until a total of 998.