[Reviews · NeurIPS 2020]

Review 1

Summary and Contributions: The paper proposes using sparse discrete distributions in problems where objective contains full sum over the discrete distribution. The imposed sparsity allows exact marginalization and deterministic gradients. The authors use top-k sparsemax, and sparseMAP to represent sparse parametric distributions. Experiments include semisupervised VAE with categorical class variable, emergent communication game and VAE with Bernoulli latent variables.

Strengths: The idea of imposing sparsity on parametric distribution in cases where the real distribution is sparse sounds useful. In particular for categorical distribution in the setting where training converges to single or few choices, this approach can potentially bring significant compute savings. The idea seems novel to me.

Weaknesses: The main weakness is the issue of scaling this approach to a setting with a large number of categories, or large number of binary variables. 1. The experiment on VAE with binary latents shows that if discrete distribution has large support than the proposed method isn't very useful. 2. Another situation where I suspect the method won't work very well is when learning distribution over a very large number of classes: early in training this categorical distribution will have full support and collapse only closer to the end of training. In this case it's possible that other baselines will perform better, because compute budget will be dominated by the early part of training and will be >>1. At the same time using top-k sparsemax might hurt the learning process again in comparison to baselines. 3. Comparison to existing baselines is very sparse: given the generality of the claims in the paper it would be useful to compare with more available baselines (e.g. REBAR, ARM)

Correctness: I find the methods used in the paper appropriate for supporting the claims, however it seems that the proposed method might have scalability issues, which should be added to the main claims in the abstract and throughout the paper.

Clarity: The paper is well written.

Relation to Prior Work: The proposed method differs from previous approaches in application of sparse discrete distribution, at least as far as I know, this distinction is clearly stated in the paper.

Reproducibility: Yes

Additional Feedback: The paper will be much stronger if the authors could include analysis of their method in cases with large number of categories/binaries. Also plotting training objectives as a function of computational resources would be useful in understanding the trade-offs. Finally adding standard baselines(Gumbel, REBAR, ARM, etc) to all experiments, including bit-vector VAE, would be helpful. ****************************post rebuttal**************** I'd like to thank authors for addressing the above concerns, I have updated my score.


Review 2

Summary and Contributions: I read the rebuttal and I don't think these questions are well-explained, probably due to the space limit. I hope the authors will keep their promise and make major updates to the paper in the final version. ------------------------------- update ----------------------------------------- This work proposes a new method to solve the gradient estimation problem of marginalization over discrete latent variables. The idea is to use sparsemax (a sparse alternative to softmax which serves as a differentiable relaxation of argmax) to define the distribution over discrete configurations, so that marginalization can be efficiently carried out given only a small number of configurations have non-zero probability. To support combinatorial latent variables where plain sparsemax is computationally infeasible. The authors propose two principled modifications: 1) top-k sparsemax by restricting the feasible set of sparsemax to have maximally k non-zero probabilities. 2) sparseMAP that finds sparse solution over combinatorial structures through an active set algorithm. The two methods are evaluated against several baselines on 1) a semi-supervised VAE on MNIST; 2) an emergent communication game; and 3) a bernoulli latent VAE.

Strengths: This is the kind of work that I have been waiting to see for some time. The sparsemax and sparseMAP are very beautiful tools and has the potential to address many combinatorial sum problems in differentiable programming, with or without structure. These methods are well-recognized in the structured prediction communities while remaining under-explored in gradient estimation literature. This paper makes timely contribution to the gradient estimation literature by exploring such ideas. Although the idea of exact marginalization under sparsity is straightforward to apply to discrete gradient estimators, the execution of it has many challenges, which is made clear by the authors in the paper. Specifically, as pointed out in L139, ``solving the problem in Eq. 2 still requires explicit manipulation of the large vector s \in R^|Z|, and even if we could avoid this, in the worst case (s = 0) the resulting sparsemax distribution would still have exponentially large support.'' The two solutions to this challenge is technically sound. I particularly like the top-k sparsemax formulation, which unifies the constraint into the feasible set (instead of using post-hoc truncation) and remains differentiable using results from sparse projection onto simplex. The sparseMAP algorithm is more complex than top-k sparsemax and it has been a question for me that what the solutions are like when applying sparseMAP to multiple independent Bernoulli latents (as in section 5.3).

Weaknesses: * I read the sparseMAP paper some time ago and I remember that the implementation details of the active set algorithm is a bit unclear to me. I'd appreciate a more detailed discussion (ideally pseudo code) in the future version of the paper, especially when it is used in gradient estimators for multiple **independent** discrete latent variables. * How does the q distribution with sparseMAP look like in the bernoulli latent VAE experiment (section 5.3), where the structure is essentially "independence"? How sparse are the optimal sparseMAP solutions in this case? Are they the same as the solutions obtained by the active set algorithm (which is guaranteed to be sparse)? * Because the solution of the sparseMAP largely depends on the active set algorithm, it would be helpful to demonstrate such solutions in real examples. * The experiment will be more convincing by comparing sparsemax/sparseMAP to advanced gradient estimation approaches such as VIMCO (multi-sample), Gumbel softmax, REBAR, ARM, direct loss minimization, etc. in the Bernoulli latent VAE experiment (sec 5.3).

Correctness: Yes.

Clarity: Yes. The clarity can be further improved with more details on the active set algorithm used in sparseMAP.

Relation to Prior Work: * As far as I know, the idea has not been explored in any published work or preprint on gradient estimators for latent variable models. * Some related work that also borrows the idea from structured prediction (e.g., direct loss minimization) to apply to gradient estimators is missing: Direct Optimization through $\arg\max $ for Discrete Variational Auto-Encoders. * The top-k sparsemax formulation is related to A Truncated EM Approach for Spike-and-Slab Sparse Coding. Though the approach taken by the submission here is considerably better (it is unified into the feasible set of sparsemax and remains differentiable).

Reproducibility: Yes

Additional Feedback: Please see the above suggestions. I'm willing to raise the rating if the questions are well-addressed.


Review 3

Summary and Contributions: The paper addresses the problem of training latent variable models with discrete latent variables. Current methods either marginalize over latent variables explicitly (which becomes intractable quite fast), or use biased or unbiased gradient estimators. The paper tackles the problem in a different way, by using sparse discrete latent variables. It does this by using sparse k-projections onto the simplex of appropriate dimension. With the sparse representation with low k, marginalization can be done efficiently and exactly (although forcing k-sparse representations lead to efficient marginalization, it may be suboptimal).

Strengths: Novelty: Sparse projections onto the simplex were previously studied in the paper by Kyrillidis et al (2013). This work present a novel way to apply these methods for discrete latent variable models, which appears to work quite well in practice. Relevance and significance: I think this paper is definitely relevant to the NeurIPS community, since new methods to train discrete latent variable models are continuously being developed, and there are several applications that use this kind of models.

Weaknesses: I have some comments regarding the empirical evaluation: 1- Gumbel-softmax has a temperature parameters that might have a significant effect on the method's performance. For the second set of experiments the appendix states that a temperature of 1 was used. Were other values tested? Was this value used for the semisupervised VAE too? 2- Other baselines that have been observed to perform well could be included. For instance, VIMCO ("Variational Inference for Monte Carlo Objectives" by Mnih et al), and Rebar ("REBAR: Low-variance, unbiased gradient estimates for discrete latent variable models" by Tucker et al).

Correctness: They appear to be correct.

Clarity: Yes.

Relation to Prior Work: Yes.

Reproducibility: Yes

Additional Feedback: *** After rebuttal *** I will maintain my score. I like the idea, it proposes a different way of dealing with discrete variables, and results look promising. *** *** - I think ELBO vs epoch plots would be a nice to have for the last set of experiments. Why were these not included? I usually find them quite informative and straightforward to interpret.


Review 4

Summary and Contributions: The paper introduces the idea of using sparse distributions in latent variable models to efficiently perform exact marginalizations of discrete variables. This is achieved by using the sparsemax operator (or its variants) instead of the usual softmax.The benefits of this approach are shown in 3 different settings, where the introduced method achieves similar results to dense marginalization while needing a much lower number of loss evaluations.

Strengths: Being able to use discrete variables in deep latent variable models is a fundamental yet challenging task. The main issue lies in the fact that exact marginalization is often intractable, and the approximations commonly used (e.g. score function estimator, continuous relaxations like Gumbel-Softmax) are practically difficult to get to work consistently. This paper introduces the novel idea of using sparse distributions over the discrete variables to solve this issue. In this way in fact it is computationally feasible to perform exact marginalization, since only a small number of terms will be non-zero (therefore greatly reducing the number of loss evaluations needed). The method is sound and fairly easy to implement, so I believe it could have an important impact in the community.

Weaknesses: The introduced method relies on sparse distributions, which is a quite strong assumption. While the authors address the main implications of this assumption, I think there should have been an even more detailed discussion/empirical evaluation to increase the impact of this work in the community: - in the semi-supervised learning experiments in section 5.1 you use a VAE model which is relatively simple and by now quite outdated. Do you expect these results to generalize to more complex architectures? For example, if I took any of the SOTA semi-supervised deep generative models and just replaced the softmax with the sparsemax would you expect similar improvements? - how does this method behave with challenging tasks that may contain many ambiguous data points? Would the model just use lots of loss evaluations throughout the whole training procedure (and not only in the beginning as in your experiments) or would the sparsity assumption make the model learn to be certain even for ambiguous data? - since sparsemax is such a core component of this method, it would be useful to add some details on its forward/backward passes and their computational complexity wrt the softmax.

Correctness: Yes.

Clarity: Yes, I enjoyed reading it

Relation to Prior Work: Yes.

Reproducibility: Yes

Additional Feedback: *** reply to author feedback *** Thanks for your rebuttal. After reading it I still argue for acceptance, since I believe that this relatively simple idea could have a good impact in the community.

[Author Response · NeurIPS 2020]

We thank all reviewers for their comments. A common suggestion, which we will follow, was comparing against more baselines. In §5.1, we compare against Sum&Sample, which was shown in prior work [Fig. 4 of Liu et al., 2019] to outperform RELAX (of which REBAR is a special case [Grathwohl et al., 2018]). During the rebuttal, we ran preliminary experiments with VIMCO which does not seem to outperform moving average baseline on the bit-vector experiment (even after searching across learning rates and number of $K$ samples; though more seeds are needed to confirm this). Upon acceptance we will normalize these baselines for all experiments and include suggested ones.

**Reviewer #1**

> *"The main weakness is the issue of scaling (...) to a setting with a large number of categories or binary variables"*

Thanks for bringing this up, scalability is an important point that we want to make sure is clear in the final version. Our goal in experimenting with latent sizes $K \in \{10, 256, 2^{32}, 2^{128}\}$ is precisely to analyze how our methods scale with $K$.

For the VAE with binary latents, top-$k$ sparsemax is indeed not the best choice when $D = 128$, since it always requires $k$ decoder calls, as shown in Fig. 3. Note, however, that our other proposed method (SparseMAP) scales well, reaching very sparse solutions from the beginning of training. This is expected, as SparseMAP is better tailored for combinatorial problems, as discussed in the *Results and discussion* paragraph of §5.3. We will make this clearer.

In the categorical case, where the number of classes $K$ can be large but not combinatorial, sparsemax appears to scale well: Fig 1 (right) and Table 1 shows that sparsemax has much fewer decoder calls than $K$; for extremely large $K$, top-$k$ sparsemax with $k \leq K$ can be used to upper bound this number in the beginning of training.

> *"The paper will be much stronger if the authors could include analysis of their method in cases with large number of categories/binaries (...) Plotting training objectives as a function of computational resources would be useful"*

We will include this analysis and plots as suggested. Thank you!

**Reviewer #2**

> *"Implementation details of the active set algorithm is a bit unclear to me. I'd appreciate a more detailed discussion"*

Good suggestion, we will add a detailed discussion and pseudo-code.

> *" How sparse are the optimal sparseMAP solutions (...)? Are they the same [solutions of] the active set algorithm? "*

The number of decoder calls on the $y$-axis of Fig.3 corresponds to the SparseMAP support. Upon convergence, the support is 1 for $> 50\%$ of the examples. In general, SparseMAP is guaranteed to have a solution spanned by $D - 1$ configurations. With enough active set iterations, this guarantees the exact solution. We set the maximum number of iterations to 300 ($\gg D - 1$), and most of the time an exact solution was found in much fewer iterations.

Thank you for sharing the missing related work, we'll include it in the final version of the paper!

**Reviewer #3**

> *"For the 2nd set of experiments (...) a temperature of 1 was used [for Gumbel-Softmax]. Were other values tested?"*

Thank you for noticing this omission. Following Jang et al. [2017], we decayed the temperature throughout training. We tuned this decay rate for both experiments §5.1 and §5.2. We'll clear this up in the final version.

We will also include ELBO vs. epoch plots in the camera-ready. Thank you for your suggestions!

**Reviewer #4**

> *"In section 5.1 (...) Do you expect these results to generalize to more complex architectures?"*

Modifications to Kingma's original semi-supervised VAE include changes to the probabilistic model (e.g. more variables, different conditional independent assumptions), architecture (e.g. increasing capacity), and objective (e.g. promoting representation orthogonality). These changes seem motivated more from the point of view of disentanglement than from a potential limitation imposed by noisy gradients. While assessing our techniques in those settings is surely interesting, we feel it lies a bit beyond our current submission.

> *"How does this method behave with challenging tasks that may contain many ambiguous data points?"*

Good question. In Figure 3 of our paper, you can catch a glimpse of this—when the bit-vector dimensionality is $D = 128$, top-$k$ sparsemax has full support throughout training, and SparseMAP is a better choice (see answer to R1). In the extreme case where all data points are genuinely ambiguous, the sparsity assumption may not be suitable.

> *"It would be useful to add some details on [sparsemax] forward/backward passes and their computational complexity"*

We will include these details, as suggested. Thank you!

[Meta-Review · NeurIPS 2020]

This paper proposes an approach for discrete latent variable models that uses sparsity in order to explicitly marginalize. The idea in this paper is simple, elegant, and a refreshing departure from the typical relaxation / reparameterization approaches to this problem. Please include the clarifications requested by the reviewers.